# Maternal Serum 25-Hydroxyvitamin D as a Possible Modulator of Fetal Adiposity: A Prospective Longitudinal Study

**DOI:** 10.3390/ijms26094435

**Published:** 2025-05-07

**Authors:** Keisuke Akita, Satoru Ikenoue, Junko Tamai, Toshimitsu Otani, Marie Fukutake, Yoshifumi Kasuga, Mamoru Tanaka

**Affiliations:** Department of Obstetrics and Gynecology, Keio University School of Medicine, Shinanomachi, Shinjuku-ku 160-8582, Tokyo, Japan

**Keywords:** fetal ultrasonography, 25-hydroxyvitamin D, fetal adiposity, gestational diabetes mellitus

## Abstract

25-hydroxyvitamin D (25(OH)D) regulates lipid metabolism, and its decrease is proposed as a pathogenesis of metabolic syndrome, gestational diabetes mellitus (GDM), and eventually fetal adiposity. Decreased 25(OH)D is also linked with the development of gestational diabetes mellitus (GDM), which is associated with increased fetal adiposity. Fetuses are dependent on the supply of 25(OH)D from maternal circulation. However, the influence of maternal serum 25(OH)D on fetal adiposity remains unclear. This study aimed to investigate the association between maternal serum 25(OH)D and fetal adiposity. A prospective longitudinal study was conducted in a cohort of 89 (including 21 GDM) singleton pregnancies. Maternal blood samples were obtained at 10, 24, 30, and 36 weeks, and fetal ultrasonography was performed at 24, 30, and 36 weeks of gestation. Estimated fetal adiposity (EFA) was calculated as the average z-score of cross-sectional arm and thigh percentage fat area and anterior abdominal wall thickness as previously reported. The multiple linear regression analyses indicated that maternal 25(OH)D levels across gestation were not associated with EFA at 24 and 30 weeks, while maternal 25(OH)D at 24 weeks was inversely correlated with EFA at 36 weeks. Particularly, in the GDM group, maternal 25(OH)D levels at 10, 24, 30, and 36 weeks all showed a significant negative correlation with EFA at 36 weeks. Decreased maternal serum 25(OH)D level could be an early biomarker of increased fetal adiposity in late gestation, especially in diabetic pregnancies.

## 1. Introduction

Intrauterine fetal growth significantly affects not only perinatal outcomes but also the early onset of metabolic disorders [1,2]. Notably, the amount of fetal fat deposition varies widely among individuals and is believed to be closely related to the incidence of childhood obesity and later metabolic dysfunction [3]. Since established obesity is difficult to reverse, primary prevention of childhood obesity is crucial [4]. For the past 50 years, estimated fetal weight has been used as the gold standard for evaluating fetal growth and predicting birth weight. It was calculated using the biparietal diameter, abdominal circumference, and femoral length [5]. While abdominal circumference reflects the amount of surrounding subcutaneous tissue, biparietal diameter, and femur length are bone components and are unaffected by soft tissue volume. Moreover, estimated fetal weight has an error margin of approximately 10% [6]. Moreover, birth weight correlates only moderately with infant percentage body fat [6].

Measuring the fat area at multiple body sites is considered a useful approach for estimating fetal fat mass [7,8]. We previously proposed the estimated fetal adiposity (EFA), a robust estimate of fetal total fat mass based on fat mass in the upper arm, thigh, and abdominal wall thickness [7]. Previous reports have suggested that the EFA improves the accuracy of the prediction of fetal adiposity since fat mass is measured at multiple anatomical sites. Additionally, fat mass calculated by EFA is significantly correlated with the body fat percentage of newborns [7].

Vitamin D is a fat-soluble vitamin primarily involved in bone metabolism and calcium homeostasis [9]. Recent research has also highlighted its involvement in lipid metabolism. Vitamin D levels are associated with body mass index (BMI) [10], with studies reporting that individuals with a high BMI have lower blood vitamin D levels and increased fat mass [10,11,12,13]. Additionally, serum vitamin D levels are associated with blood cholesterol and fatty acid levels [14]. Animal experiments have shown that adding vitamin D to the diet reduces body fat [15].

Women of reproductive age are particularly susceptible to vitamin D insufficiency [16,17]. Vitamin D insufficiency or deficiency is associated with adverse perinatal outcomes, including preeclampsia, gestational diabetes mellitus (GDM), cesarean section, premature birth, and low birth weight [18,19]. Previous studies have shown that maternal vitamin D deficiency is associated with estimated fetal weight or offspring obesity [20,21,22], but no studies have examined the relationship between maternal vitamin D levels and fetal adiposity.

GDM is one of the most common perinatal complications. Infants born to mothers with GDM have a higher incidence of macrosomia owing to increased fetal fat deposition, especially in the trunk and upper arm [8]. Notably, vitamin D has the effect of improving insulin resistance, suggesting its potential to prevent GDM and improve glycemic control during pregnancy [23]. Although the effects of vitamin D during pregnancy are increasingly recognized, its impact on fetal growth remains unclear. Specifically, the relationship between fetal body composition or fat mass and maternal serum vitamin D levels has not been described.

Therefore, this study aimed to clarify the relationship between maternal serum vitamin D levels and fetal adiposity during gestation. Since fetal fat deposition accelerates in the third trimester and is affected by maternal insulin resistance [8,24], we hypothesized that maternal serum vitamin D levels affect fetal fat mass in late gestation, particularly in pregnancies complicated by GDM.

## 2. Results

Maternal characteristics and perinatal outcomes are presented in Table 1.

A total of 21 (23.6%) women were diagnosed with GDM. Maternal serum 25-hydroxyvitamin D (25(OH)D) levels at 10, 24, 30, and 36 weeks of gestation were 18.84 ng/mL, 16.46 ng/mL, 18.02 ng/mL, and 16.41 ng/mL, respectively, with no significant differences observed between the GDM and non-GDM groups at any time point. Additionally, seasonal variations (spring, summer, autumn, and winter) did not significantly affect 25(OH)D levels. The fetal ultrasound measurements are shown in Table 2. EFA results did not differ significantly between the GDM and non-GDM groups.

Maternal serum 25(OH)D levels at all time points did not correlate with EFA levels at 24 and 30 weeks. However, maternal serum 25(OH)D levels at 24 weeks negatively correlated with EFA at 36 weeks, while 25(OH)D levels at 10, 30, and 36 weeks did not correlate with EFA at 36 weeks (Table 3) (Figure 1).

Sensitivity analyses showed no significant associations between maternal serum 25(OH)D levels at any time point and EFA at 24 and 30 weeks in both the GDM and non-GDM groups, as with the total cohort. However, in the GDM group, 25(OH)D levels at all time points were significantly associated with EFA at 36 weeks (Figure 2).

In the covariate analysis, maternal serum 25(OH)D levels at 10 (*p* = 0.003) and 30 weeks (*p* = 0.035) were significantly higher in women who conceived naturally than in those who underwent in vitro fertilization. Additionally, EFA at 36 weeks was associated with maternal age (*r* = −0.246, *p* = 0.020) and was significantly higher in female fetuses compared with male fetuses (*p* = 0.035). Maternal pregravid BMI and gestational weight gain did not correlate with maternal serum 25(OH)D levels or EFA. Based on these results, maternal age, method of conception, and fetal sex were identified as confounding factors for the analysis. Multiple regression analyses adjusted for these covariates indicated that maternal serum 25(OH)D levels at 10, 24, and 30 weeks were significantly associated with EFA levels at 36 weeks in the GDM group (Table 4).

Maternal 25(OH)D levels at 36 weeks showed a close (but insignificant) correlation with EFA levels at 36 weeks. In the total cohort and non-GDM group, maternal serum 25(OH)D levels at all time points were not associated with the EFA levels at 36 weeks. Additionally, maternal serum 25(OH)D levels during gestation were not correlated with birth weight percentiles or neonatal height.

## 3. Discussion

To our knowledge, this is the first study to investigate the relationship between maternal serum 25(OH)D levels and fetal adiposity during gestation. Our results showed that maternal serum 25(OH)D levels negatively correlated with EFA levels at 36 weeks, especially in mothers with GDM.

Studies in adults have shown that low vitamin D levels lead to higher BMI and body fat percentage [10]. However, no prior reports have explored the relationship between maternal serum vitamin D levels and fetal fat mass. While several reports have examined maternal vitamin D deficiency during pregnancy and the body composition of offspring at 3–6 years of age, their interpretation remains controversial [25,26,27]. Some studies have reported that children born to vitamin D-deficient mothers have an increased body fat percentage during childhood [25,28]. In contrast, others have found no correlation between maternal vitamin D levels and childhood obesity [26,29]. This study revealed the relationship between maternal vitamin D levels and fetal fat mass, suggesting that it may influence body composition from birth to childhood.

As discussed above, there appears to be a relationship between vitamin D levels and fetal adiposity, with serum vitamin D involved in lipid metabolism through multiple mechanisms [14]. First, vitamin D inhibits the action of sterol regulatory element-binding proteins, which are necessary for the synthesis and absorption of cholesterol and other fatty acids in the body [30]. Second, increased calcium absorption in the intestinal tract by serum 25(OH)D reduces triglyceride secretion and synthesis in the liver [31]. Third, the inhibition of parathyroid hormone function by vitamin D suppresses lipogenesis and reduces serum triglyceride levels [32]. Conversely, 1,25(OH)2D is associated with obesity [33]. In molecular biology, 1,25(OH)2D regulates the development and differentiation of adipocytes and alleviates obesity by inhibiting the expression of peroxisome proliferator-activated receptor γ (PPARγ). Patients with obesity also have higher levels of vitamin D-binding protein, which reduces the circulating levels of free 25(OH)D. Furthermore, placental 1,25(OH)2D 24-hydroxylase (CYP24A1), which may help to break down 25(OH)D and 1,25(OH)2D into inactive metabolites, is strongly expressed in women with GDM, resulting in lower vitamin D levels than in normal pregnant women.

Several studies using laboratory animals have reported on the relationship between vitamin D and obesity. Supplementation with 1,25(OH)2D3 in mice prevents diet-induced obesity, improves glucose homeostasis and lipid oxidation, suppresses weight gain and lipogenesis, and promotes fatty acid oxidation [15,34]. However, the relationship between vitamin D and fetal fat metabolism remains unclear. Vitamin D in maternal serum crosses the placenta and is transferred to the fetus [19]. Considering that a similar mechanism to that in adults occurs in the fetus, maternal vitamin D levels might influence fetal fat metabolism. These findings suggest that a relationship exists between vitamin D and lipid metabolism and that vitamin D deficiency increases serum lipid levels. Furthermore, lipid components, including free fatty acids, may cross the placenta and reach the fetus, thereby affecting fetal body composition.

In the GDM group, maternal serum Vitamin D levels at all gestational ages were significantly negatively correlated with EFA at 36 weeks of pregnancy. A significant correlation was also observed in the multivariate analysis, especially during early to mid-pregnancy. The relationship between vitamin D levels and fetal fat mass was more pronounced in the GDM group than in the non-GDM group. Pregnant women with GDM have higher fetal birth weights. They have a higher limb volume during the fetal period than normal pregnant women, likely owing to increased body fat mass in pregnant women with GDM [24]. The number of FFAs that cross the placenta increases in pregnant women with GDM. Triglycerides are hydrolyzed to free fatty acids, and a significant positive correlation exists between serum triglyceride levels and birth weight. Additionally, insulin resistance during pregnancy correlates with fetal fat mass [8]. In pregnant women with GDM, increased insulin resistance and fatty acids, such as triglycerides, are suggested to contribute to increased fat mass in newborns.

One of the effects of vitamin D is improving insulin resistance [23]. Vitamin D deficiency reduces insulin sensitivity, increases triglyceride and low-density lipoprotein cholesterol production, and reduces high-density lipoprotein cholesterol synthesis. 1,25(OH)2D regulates insulin synthesis and secretion by binding to vitamin D receptors in pancreatic islet cells. First, it controls the intracellular and extracellular calcium balance by enhancing the function of voltage-gated calcium channels and increasing calcium influx [35]. Second, it stimulates calcium-dependent insulin synthesis, leading to the mobilization of secretory vesicles, including insulin granules [36]. Additionally, vitamin D is involved in the expression of many genes, including those that promote glucose-stimulated insulin secretion. By upregulating the expression of related genes, vitamin D promotes glucose-stimulated insulin secretion [37]. By binding to vitamin D receptors in peripheral tissues, 1,25(OH)2D stimulates insulin receptor expression and promotes insulin-mediated glucose transport, thereby directly improving insulin sensitivity [36]. Furthermore, it transforms T helper cells into an anti-inflammatory Th-2 subset, suppressing inflammatory responses in the placenta and pancreatic islets and reducing insulin resistance [38].

When pregnant women with GDM are supplemented with vitamin D, fasting blood glucose levels, insulin concentrations, and insulin resistance are significantly reduced [18]. A Cochrane review also found that vitamin D supplementation reduces the risk of developing GDM [39]. Based on this finding, increasing maternal vitamin D levels is expected to reduce fetal fat mass by improving maternal insulin resistance.

In this study, we identified pregnancy type as a factor related to maternal serum 25(OH)D levels and included it as a confounding factor in our analyses. In the perinatal field, vitamin D may be involved in the implantation of a fertilized egg [19]. Active vitamin D improves sperm–egg binding by increasing intracellular calcium levels and acrosin activity. This enzyme breaks down the outer wall of the egg, allowing sperm to penetrate the egg and complete the fertilization process [40]. Additionally, a study found that male rats with vitamin D deficiency had a 44% lower reproductive success rate than their vitamin D-sufficient counterparts [41]. Consequently, pregnant women who conceived through in vitro fertilization in this study possibly had lower serum 25(OH)D levels.

Maternal age and fetal sex are factors influencing fetal EFA. Women have more fat mass than men, and this has been observed immediately after birth and several months later [42]. Additionally, female fetuses exhibit higher fat mass compared to males at 30–36 weeks of pregnancy, as measured by MRI [43], which does not contradict the findings of this study. Although no previous reports linking maternal age to fetal fat have been published, the probability of intrauterine growth retardation and low birth weight increases with age. The results of this study may reflect this relationship [44].

A notable strength of our study lies in the prospective design of fetal ultrasound examinations, minimizing selection bias. Additionally, the sonographers performing the ultrasounds were blinded to the serum vitamin D levels, ensuring unbiased measurements. However, certain limitations should be acknowledged. The patient’s background may have influenced the findings. In previous reports, vitamin D was related to BMI [10]; however, we found no correlation between maternal BMI and serum vitamin D levels. Only approximately 7% of pregnant women in this study had obesity (which is common in Japanese women), which may have affected the results. These differences in background characteristics may have impacted the relationship between maternal BMI and serum vitamin D levels. Furthermore, this study did not assess vitamin D intake among the participants. Although maternal vitamin D levels are essential for placental transfer to the fetus, this limitation likely had a minimal impact on the study’s findings. Another limitation of this study is that it was conducted at a single facility, and the number of study subjects was relatively small. It is possible that altered findings could be obtained by expanding the study subjects in the future.

## 4. Materials and Methods

This prospective longitudinal study included 89 women with singleton pregnancies recruited during the first trimester between November 2021 and December 2023. Patients with pre-gestational diabetes, chronic hypertension, fetal structural malformations, chromosomal abnormalities, or fetal hydrops were excluded. The study was approved by the Institutional Review Board (#20210011, approved on 11 March 2021), and written informed consent was obtained from all participants.

Maternal blood samples were collected at 10, 24, 30, and 36 weeks of gestation. Blood samples were centrifuged at 3000 rpm for 10 min, and the serum was stored at −80 °C. The 25(OH)D levels were measured using a chemiluminescent enzyme immunoassay, with an intra-assay coefficient of variation of 5.59%.

Fetal ultrasonography was performed at approximately 24, 30, and 36 weeks of gestation to assess fetal biometry. Mid-trimester blood tests and fetal ultrasonography were performed at 24 weeks when the screening for gestational diabetes was performed. Then, we performed a maternal blood test and fetal ultrasonography every 6 weeks (30 and 36 weeks of gestation). Each conventional fetal biometric measurement was obtained in duplicate and averaged. All fetal measurements were performed by an obstetrician (J.T.) using Voluson E10 (GE Healthcare, Chicago, IL, USA) with a matrix array transducer (RM6C). Measurements were adjusted for gestational age on an ultrasound scan, as previously reported [7,8]. Estimated fetal weight was calculated using the commonly used model described by the Japan Society of Ultrasound in Medicine [45]. Fetal fat mass was measured using 4D View 9.0 (GE Healthcare, Milwaukee, WI, USA) to determine the subcutaneous fat area from standardized cross-sectional images. The fat area was calculated as the difference between the total cross-sectional area and the lean mass area. Fetal subcutaneous fat thickness in the abdomen was measured as the high-echoic region at the traditional abdominal circumference view, 2–3 cm lateral to the cord insertion. EFA was calculated by integrating measurements of the cross-sectional upper arm and thigh percentage fat area with anterior abdominal wall thickness, as previously reported [7]. These three anatomic sites were chosen because of their reliability for subcutaneous fat assessment [7]. Because these parameters have different units of measurement, we first converted each measurement unit into standardized z-scores based on our study population. We subsequently computed the mean (unweighted) of the three z-scores as a composite estimate of fetal adiposity. The intra-observer coefficients of variation for arm percent fat area, thigh percentage fat area, and anterior abdominal wall thickness were 7.6%, 6.8%, and 6.0%, respectively.

GDM was diagnosed using a 75 g oral glucose tolerance test based on the clinical recommendations of the Japan Society of Obstetrics and Gynecology [46]. GDM was diagnosed if one or more of the following thresholds were exceeded: 92 mg/dL during fasting, 180 mg/dL at 1 h after glucose loading, or 153 mg/dL at 2 h after glucose loading. Plasma glucose levels were measured using the glucose oxidase method and enzyme immunoassay.

Maternal background data (pre-pregnancy height, weight, BMI, method of pregnancy, and previous history) and newborn outcomes (gestational age at birth, birth weight, and sex) were retrieved from medical records. Gestational age was calculated based on the last menstrual period and confirmed by first-trimester ultrasound. Birth weight was converted to birth weight percentiles using the Japanese standard sex- and parity-specific birth weight percentile charts [47].

Statistical analyses were performed using univariate analyses of maternal serum vitamin D levels and EFA across gestation periods. Multivariate analyses were subsequently performed to account for confounding factors. The confounding factors identified were maternal age, pre-pregnancy BMI, method of pregnancy, gestational weight gain, and fetal sex, which were associated with maternal serum vitamin D levels or EFA. Finally, sensitivity analyses were conducted by stratifying the cohort into the GDM and non-GDM groups. All statistical analyses were performed using SPSS software (version 29.0; SPSS Inc., Chicago, IL, USA), with statistical significance set at *p* < 0.05.

## 5. Conclusions

Maternal serum 25(OH)D level was inversely associated with fetal adiposity in late gestation. Maternal 25(OH)D could be an early biomarker of fetal adiposity, especially in diabetic pregnancies. Further investigations for increasing serum 25(OH)D levels in mothers with GDM (through diet or supplements) might reduce fetal adiposity, which potentially leads to the primary prevention of increased newborn adiposity and subsequent childhood obesity and early onset metabolic syndrome in later life.

## Figures and Tables

**Figure 1 ijms-26-04435-f001:**
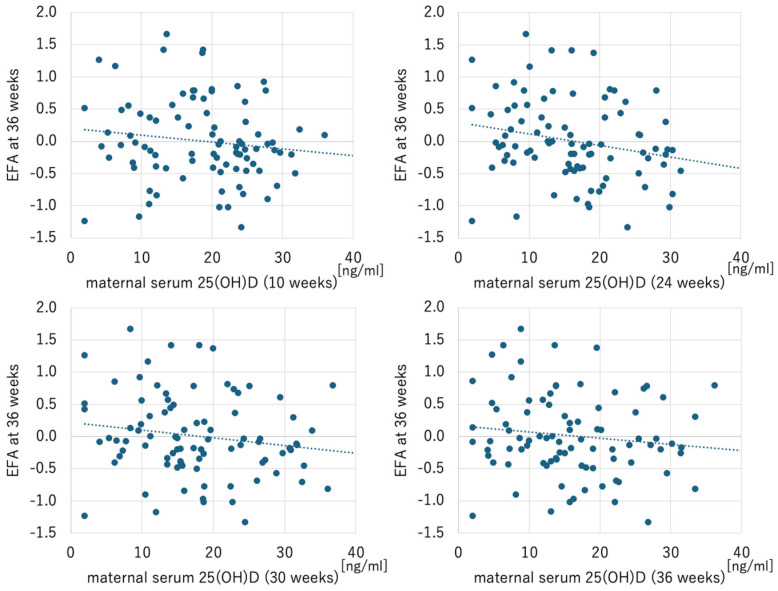
Scatterplots of maternal serum 25-hycroxyvitamin D and EFA at 36 weeks. Maternal serum 25-hydroxyvitamin D at 24 weeks (but not at 10, 30, and 36 weeks) was significantly correlated with EFA at 36 weeks (*r* = −0.234, *p* = 0.028). EFA, estimated fetal adiposity.

**Figure 2 ijms-26-04435-f002:**
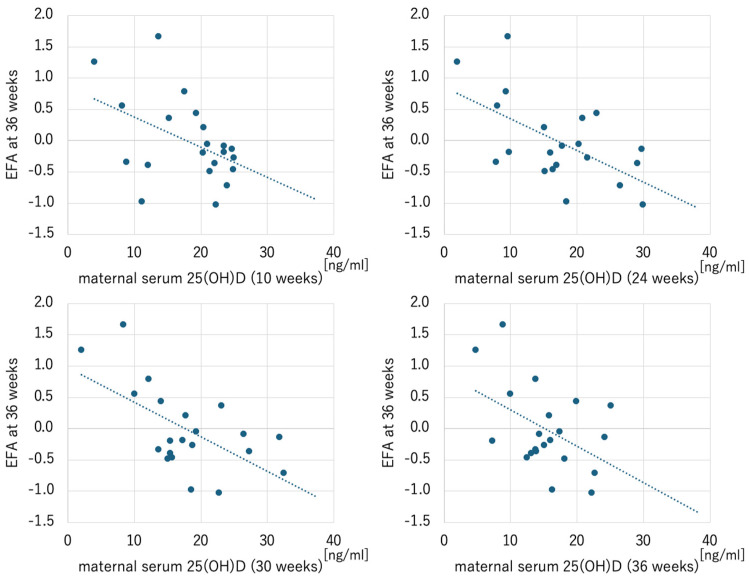
Scatterplots of maternal serum 25-hydroxyvitamin D and EFA at 36 weeks in the GDM group. Maternal serum 25-hydroxyvitamin D at 10, 24, 30, and 36 weeks were significantly correlated with EFA at 36 weeks (10 weeks: *r* = −0.450, *p* = 0.041; 24 weeks: *r* = −0.585, *p* = 0.005; 30 weeks: *r* = −0.607, *p* = 0.004; 36 weeks: *r* = −0.461, *p* = 0.035). EFA, estimated fetal adiposity.

**Table 1 ijms-26-04435-t001:** Maternal and neonatal characteristics (*n* = 89).

Characteristics	Mean ± SD or N (%)
Age, years	34.9 ± 3.9
Pre-pregnancy BMI, kg/m^2^	20.9 ± 2.7
Primiparous	55 (62%)
Gestational weight gain, kg	10.5 ± 3.6
Gestational diabetes	21 (24%)
Hypertensive disorders in pregnancy	6 (6.7%)
Birth weight, g	3026 ± 330
Birth weight percentile, %	58.9 ± 26.5
Gestational age at birth, week	38.9 ± 1.0
Infant sex (female)	43 (48%)
Maternal serum 25-hydroxyvitamin D, ng/mL	
at 10 weeks	18.8 ± 8.2
at 24 weeks	16.5 ± 8.3
at 30 weeks	18.0 ± 9.2
at 36 weeks	16.4 ± 9.2

Data are presented as mean ± SD or N (%).

**Table 2 ijms-26-04435-t002:** Fetal biometry measured by ultrasonography (n = 89).

Parameters	24 Weeks	30 Weeks	36 Weeks
Estimated fetal weight, g	764 ± 153	1572 ± 153	2562 ± 247
Arm total area, mm^2^	2.84 ± 0.56	4.69 ± 0.77	7.97 ± 1.21
Arm lean area, mm^2^	1.50 ± 0.31	2.30 ± 0.39	3.47 ± 0.55
Arm fat area, mm^2^	1.33 ± 0.40	2.40 ± 0.56	4.50 ± 0.94
Thigh total area, mm^2^	5.73 ± 1.09	10.40 ± 1.51	17.36 ± 2.75
Thigh lean area, mm^2^	3.56 ± 0.64	6.04 ± 0.96	9.38 ± 1.48
Thigh fat area, mm^2^	2.17 ± 0.67	4.36 ± 0.93	7.98 ± 1.78
Anterior abdominal wall thickness, mm	2.25 ± 0.53	2.92 ± 0.61	4.04 ± 0.96
EFA	0.01 ± 0.64	0.03 ± 0.66	0.00 ± 0.64

Data are presented as mean ± SD. EFA, estimated fetal adiposity.

**Table 3 ijms-26-04435-t003:** Univariate analyses between maternal serum 25(OH)D and EFA (n = 89).

Parameters	EFA at 24 Weeks		EFA at 30 Weeks		EFA at 36 Weeks	
	r	*p*-Value	r	*p*-Value	r	*p*-Value
Maternal serum 25(OH)D						
at 10 weeks	−0.209	0.051	−0.047	0.661	−0.136	0.205
at 24 weeks	−0.019	0.861	−0.101	0.347	−0.234	0.028
at 30 weeks	−0.034	0.754	−0.138	0.197	−0.174	0.103
at 36 weeks	−0.063	0.557	−0.083	0.442	−0.137	0.200

25(OH)D, 25-hydroxyvitamin D; EFA, estimated fetal adiposity.

**Table 4 ijms-26-04435-t004:** Multiple regression model associating maternal serum 25(OH)D with EFA at 36 weeks of gestation adjusted for the confounding factors (maternal age, fetal sex, method of pregnancy).

Parameters	Total (n = 89)		GDM (n = 21)		Non-GDM (n = 68)	
	Unstandardized B [95%CI]	*p*-Value	Unstandardized B [95%CI]	*p*-Value	Unstandardized B [95%CI]	*p*-Value
Maternal serum 25(OH)D						
at 10 weeks	−0.007 [−0.024–0.009]	0.391	−0.049 [−0.088–−0.010]	0.017	−0.002 [−0.020–0.017]	0.864
at 24 weeks	−0.012 [−0.028–0.004]	0.134	−0.042 [−0.071–−0.013]	0.008	−0.005 [−0.023–0.014]	0.619
at 30 weeks	−0.007 [−0.021–0.008]	0.378	−0.045 [−0.080–−0.010]	0.014	0.000 [−0.017–0.016]	0.971
at 36 weeks	−0.005 [−0.021–0.090]	0.461	−0.044 [−0.093–0.006]	0.089	−0.002 [−0.018–0.013]	0.766

25(OH)D, 25-hydroxyvitamin D; EFA, estimated fetal adiposity; GDM, gestational diabetes mellitus; CI, confidence interval.

## Data Availability

The data that support the findings of this study are available on request from the corresponding author. The data are not publicly available due to privacy or ethical restrictions. This study was conducted using institutional data with the approval of the institutional ethics committee. The institutional ethics committee has not approved providing the data to a third party, and separate permission from the ethics committee is required for the data to be publicly shared.

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
