# Peer review of "Maternal Serum 25-Hydroxyvitamin D as a Possible Modulator of Fetal Adiposity: A Prospective Longitudinal Study"

_ijms, 2025, doi:10.3390/ijms26094435_

Round 1

Reviewer 1 Report

Comments and Suggestions for Authors

I congratulate the authors on the manuscript and their interesting study.

The abstract as well as the introduction states clear objectives of the study.

Material and methods are correct. It is only worth mentioning that, methods of measuring the fat tissue have their limitations in the case of fetuses with hydrops.

A suprising fact mentioned, by the authors is the relatively small average BMI. This may be worth expanding as a comparison group for a future study.

The study group of 89 cases is relatively small, although this does not detract from the quality of the manuscript.

I would swap the order of materials and methods with the discussion, the layout proposed by the authors makes it necessary to go back to the previous text.

The usg standard was given according to Japanese Standards. If they are in accordance with ISUOG standards I do not see a problem, but for future research it is better to use ISUOG criteria.

The discussion carried out correctly, refers to other studies in a reliable way.

The authors presented the strengths as well as the limitations of their study in a real and conscious way.

Minor language corrections are advised.

If possible, update the literature especially thoes over the last 15 years.

Author Response

Comment:

Material and methods are correct. It is only worth mentioning that, methods of measuring the fat tissue have their limitations in the case of fetuses with hydrops.

Response: Thank you for the comment. In this study, there were no cases of fetal hydrops. In addition, any cases of fetal hydrops were to be excluded from the study.

Specific page and line on which change(s) made: Page 7, Line 245

Changes:

Original: Patients with pre-gestational diabetes, chronic hypertension, fetal structural malformations, or chromosomal abnormalities were excluded.

Revised: Patients with pre-gestational diabetes, chronic hypertension, fetal structural malformations, chromosomal abnormalities, or fetal hydrops were excluded.

Comment:

A suprising fact mentioned, by the authors is the relatively small average BMI. This may be worth expanding as a comparison group for a future study.

Response: Thank you for the comment. As we increase the number of cases in the future, we would like to increase the number of cases with a higher BMI. Also, we have noted that the low BMI of the study population is one of the limitations (page7, Line 232).

Comment:

The study group of 89 cases is relatively small, although this does not detract from the quality of the manuscript.

Response: Thank you for the comment. We added other limitations that the small study population.

Specific page and line on which change(s) made: Page 7, Line 239

Added: Another limitation of this study is that it was conducted at a single facility and the number of study subjects was small (89 pregnancies). It is possible that altered findings could be obtained by expanding the study subjects in the future.

Comment:

I would swap the order of materials and methods with the discussion, the layout proposed by the authors makes it necessary to go back to the previous text.

Response: Thank you for the comment. I have created the papers in the same order as the thesis guidelines. I will change the order if necessary.

Comment:

The usg standard was given according to Japanese Standards. If they are in accordance with ISUOG standards I do not see a problem, but for future research it is better to use ISUOG criteria.

Response: Thank you for the comment. This study was conducted in Japan, so we used Japanese ultrasound standards.

Comment:

If possible, update the literature especially those over the last 15 years.

Response: Thank you for the comment. Among the reference papers, some published after 2010 have been updated to the latest versions.(page 2, Line 52)

Reviewer 2 Report

Comments and Suggestions for Authors

The present study deals with the effect of maternal 25 hydroxy vitamin D levels in serum on the emerge of fetal adiposity. This study seems to lack the required novelty and feasibility to this journal.

  • Please defend the novelty of this study as there are some published papers related to this topic
  • Maternal vitamin D status in pregnancy is associated with adiposity in the offspring: findings from the Southampton Women’s Survey (2012)
  • Deficit of vitamin D in pregnancy and growth and overweight in the offspring (2015)
  • Low maternal vitamin D status in pregnancy increases the risk of childhood obesity (2018)
  • This study lacks the molecular basis on which the authors made the conclusion, which is one of the main and core aspects of the international journal of molecular sciences.
  • The number of examined women especially the diabetic ones is too low to conclusive outcomes
  • Line 10-12: I think it is better to merge these outcomes as follow: “its decrease is proposed as a pathogenesis of metabolic syndrome, gestational diabetes mellitus (GDM), and eventually fetal adiposity.
  • Line 14: “samples were obtained” instead of “sample was obtained”.
  • Line 17-18: on what basis did the authors determine the timing of maternal blood collection and the fetal ultrasonography?

Author Response

The present study deals with the effect of maternal 25 hydroxy vitamin D levels in serum on the emerge of fetal adiposity. This study seems to lack the required novelty and feasibility to this journal.

Response: Thank you for the comment. We have revised the content based on your comments.

Comment:

Please defend the novelty of this study as there are some published papers related to this topic

Response: Thank you for the comment and present the papers. I added additional content with reference to these papers.

Specific page and line on which change(s) made: Page 2,  Line 61

Added: Previous studies have shown that maternal vitamin D deficiency is associated with estimated fetal weight or offspring obesity(20-22), but no studies have examined the relationship between maternal vitamin D levels and fetal adiposity.

Comment:

This study lacks the molecular basis on which the authors made the conclusion, which is one of the main and core aspects of the international journal of molecular sciences.

Response: Thank you for the comment. Since this is a clinical study, no molecular science findings have been identified. Relevant molecular science considerations are described in the Discussion section (page 6, line 158, page 6, line166 and page6, line 197). I revised some contents in this section.

Changes

Original: By inhibiting the expression of peroxisome proliferator-activated receptor γ (PPARγ), it regulates the development and differentiation of adipocytes and alleviates obesity.

Revised: In molecular biology, 1,25(OH)2D regulates the development and differentiation of adipocytes and alleviates obesity by inhibiting the expression of peroxisome proliferator-activated receptor γ (PPARγ). (page6, line158)

Added: Several studies using laboratory animals have reported on the relationship between vitamin D and obesity.(page 6, line166)

Original: Additionally, vitamin D promotes glucose-stimulated insulin secretion by upregulating the expression of related genes.

Revised: Additionally, vitamin D is involved in the expression of many genes, including those that promote glucose-stimulated insulin secretion. By upregulating the expression of related genes, vitamin D promotes glucose-stimulated insulin secretion.(page6, line197)

Comment:

The number of examined women especially the diabetic ones is too low to conclusive outcomes.

Response: Thank you for the comment. We added other limitations that the small study population.

Specific page and line on which change(s) made: Page 7, Line 239

Added: Another limitation of this study is that it was conducted at a single facility and the number of study subjects was relatively small. It is possible that altered findings could be obtained by expanding the study subjects in the future.

Comment:

Line 10-12: I think it is better to merge these outcomes as follow: “its decrease is proposed as a pathogenesis of metabolic syndrome, gestational diabetes mellitus (GDM), and eventually fetal adiposity.

Response: Thank you for the comment. The sentence has been modified as suggested.

Specific page and line on which change(s) made: Page 1, Line 10

Change

Original: 25-hydroxyvitamin D (25(OH)D) regulates lipid metabolism, and its decrease is proposed as a pathogenesis of metabolic syndrome. Decreased 25(OH)D is also linked with the development of gestational diabetes mellitus(GDM) which is associated with increased fetal adiposity.

Revised: 25-hydroxyvitamin D (25(OH)D) regulates lipid metabolism, and its decrease is proposed as a pathogenesis of metabolic syndrome, gestational diabetes mellitus(GDM), and eventually fetal adiposity.

Comment:

Line 14: “samples were obtained” instead of “sample was obtained”.

Response; Thank you for the comment. The sentence has been modified as suggested.

Specific page and line on which change(s) made: Page 1, Line 18

Change

Original: Maternal blood sample was  obtained at 10, 24, 30, and 36 weeks, and fetal ultrasonography was performed at 24, 30, and 36 weeks gestation.

Revised: Maternal blood samples were obtained at 10, 24, 30, and 36 weeks, and fetal ultrasonography was performed at 24, 30, and 36 weeks gestation.

Comment:

Line 17-18: on what basis did the authors determine the timing of maternal blood collection and the fetal ultrasonography?

Response: Thank you for the comment.  I added information about deciding when to perform ultrasound and blood sampling in Materials and Methods section.

Added: Mid-trimester blood test and fetal ultrasonography were performed at 24 weeks, when the screening for gestational diabetes was performed. Then we performed maternal blood test and fetal ultrasonography every 6 weeks (30 and 36 weeks of gestation). (p8, line 255)

Round 2

Reviewer 2 Report

Comments and Suggestions for Authors

the authors have addressed all the comments raised by the reviewer sufficiently.